# InterActHuman: Multi-Concept Human Animation with Layout-Aligned Audio Conditions

**Zhenzhi Wang**[*1], **Jiaqi Yang**[*2], **Jianwen Jiang**[*2✉], **Chao Liang**[2], **Gaojie Lin**[2],
**Zerong Zheng**[2], **Ceyuan Yang**[2], **Yuan Zhang**[2], **Mingyuan Gao**[2], **Dahua Lin**[1]
[1]The Chinese University of Hong Kong    [2]ByteDance

## Abstract

End-to-end human animation with rich multi-modal conditions, e.g., text, image and audio has achieved remarkable advancements in recent years. However, most existing methods could only animate a single subject and inject conditions in a global manner, ignoring scenarios where multiple concepts could appear in the same video with rich human-human interactions and human-object interactions. Such a global assumption prevents precise and per-identity control of multiple concepts including humans and objects, therefore hinders applications. In this work, we discard the single-entity assumption and introduce a novel framework that enforces strong, region-specific binding of conditions from modalities to each identity's spatiotemporal footprint. Given reference images of multiple concepts, our method could automatically infer layout information by leveraging a mask predictor to match appearance cues between the denoised video and each reference appearance. Furthermore, we inject local audio condition into its corresponding region to ensure layout-aligned modality matching in an iterative manner. This design enables the high-quality generation of human dialogue videos between two to three people or video customization from multiple reference images. Empirical results and ablation studies validate the effectiveness of our explicit layout control for multi-modal conditions compared to implicit counterparts and other existing methods. Video demos are available at `https://zhenzhiwang.github.io/interacthuman/`

## 1 Introduction

By leveraging the priors of pretrained Diffusion Transformer-based (DiT) video diffusion models (Bar-Tal et al., 2024; Blattmann et al., 2023a;b; Guo et al., 2023; Zhou et al., 2022; Gupta et al., 2023; Wang et al., 2023; Ho et al., 2022; Brooks et al., 2022; Wang et al., 2020; Singer et al., 2022; Li et al., 2018; Villegas et al., 2022; Lin et al., 2025b), end-to-end human animation models, especially audio-driven approaches (He et al., 2023; Tian et al., 2025b; Xu et al., 2024a; Wang et al., 2024a; Chen et al., 2024; Xu et al., 2024b; Stypulkowski et al., 2024; Jiang et al., 2024a; Lin et al., 2024; 2025a) have achieved high-quality human-centric video generation and strong controllability from multi-modal conditions, such as text, image and audio. However, most existing methods commonly hold an assumption of a single-identity paradigm: all available conditions should be fused globally and implicitly assumed to describe one unique subject in the given image. Although this global injection strategy simplifies conditioning by sharing the same condition signal across all regions, it fundamentally limits scalability in scenarios involving multiple individuals or complex human-object interactions, where each entity requires distinct appearance and voice attributes.

Recent multi-concept video customization methods, such as Video-Alchemist (Chen et al., 2025a), ConceptMaster (Huang et al., 2025), Phantom (Liu et al., 2025), and SkyReels-A2 (Fei et al., 2025a), enable injecting multiple reference images into a single video, facilitating multi-person or human-object interaction scenarios. However, these methods face significant challenges when directly applied to human animation tasks. In such tasks, animation signals are highly specific to individual identities and demand precise conditioning and alignment with specific spatiotemporal regions. For example, audio signals are exclusively associated with the current speaker and are unrelated to the background

---

*Equal contribution.

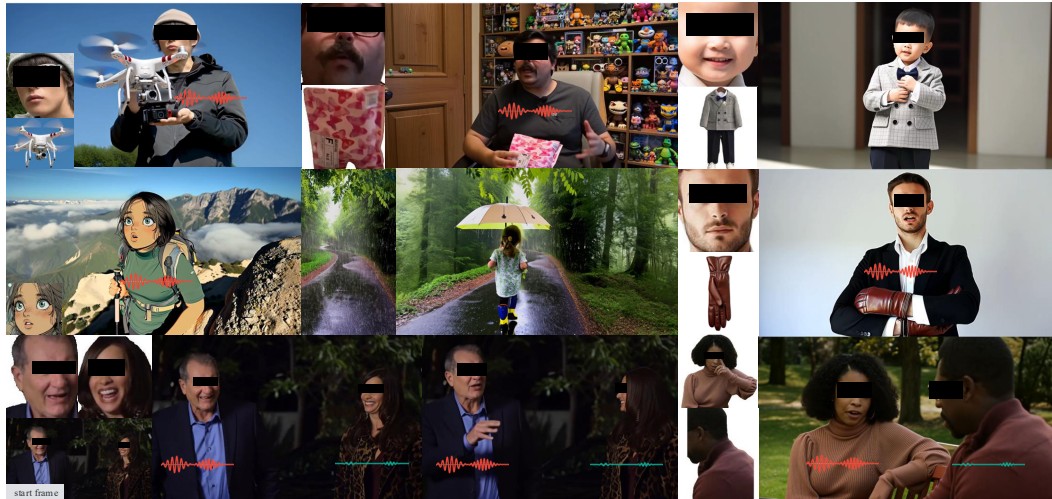

Figure 1: Video frames generated from audio and multi-concept reference images (human heads/full bodies, objects, scenes) display rich, audio-matched expressions. Our method enables compositional generation including outfit changes, human–object interactions, anime styles, dialogues even without a start frame. Red and green wave icons denote speaking and listening, respectively.

or other individuals' concepts. Existing customization methods, akin to end-to-end human animation approaches, still adopt video-level condition injection. While this approach may suffice for general video generation, it often causes confusion in human-centric video generation, making it difficult to produce satisfactory results, as shown in Fig. 3. For end-to-end human-centric video generation, conditioning inputs should ideally include not only global modalities (e.g., reference images and text descriptions) but also local modalities (e.g., audio). As mentioned, existing human animation and multi-concept customization methods fail to address this critical distinction. This limitation motivates us to propose a new framework that enables the precise injection of local human-related modalities, a capability that is both essential and urgently needed for robust multi-concept, human-centric video generation, as illustrated in Fig. 1.

In this paper, we propose **InterActHuman**, a video diffusion framework for spatially aligning multi-modal conditions in multi-concept human video generation. Unlike prior methods (Chen et al., 2025a; Huang et al., 2025) that rely on feature fusion and attention to implicitly learn relationships between condition signals and concepts, InterActHuman introduces an attention module that explicitly predicts the spatial locations where the reference concepts appear in the video. This explicit layout allows the model to accurately associate local audio conditions with the correct regions via iterative mask prediction and masked audio attention during diffusion inference. Our framework offers two key advantages. First, it enforces a stronger layout constraint by precisely binding each condition to its corresponding spatial region. Second, it provides a unified interface for synchronously injecting all modalities (e.g., visual and acoustic inputs) through the layout. These features make InterActHuman well suited for multi-modal, multi-concept human animation and establish a baseline for this domain.

Although using an explicit mask for condition injection may seem straightforward, it creates a chicken-and-egg dilemma: during inference the final video is not yet available, leaving the spatial positions of each identity uncertain and making accurate mask prediction impossible; yet without those masks, spatial injection of local audio conditions cannot be performed, leading to an incomplete or misaligned generation process. To address this, we leverage the iterative denoising process inherent in diffusion models (Song et al., 2021). Specifically, we introduce a mask-predictor branch into the diffusion pipeline and adopt an interleaved mask-prediction strategy, wherein the mask predicted at step $k$ guides condition injection at step $k+1$. This iterative refinement progressively finalizes the spatial layout, breaking the cyclic dependency and enabling precise spatial alignment even without ground-truth video during inference. In essence, our approach converts the chicken-and-egg problem into a sequential, convergent procedure that robustly aligns local audio conditions.

Beyond model design, we developed a scalable pipeline to automatically assemble high-quality, human-centric animation data to address the lack of suitable multi-concept datasets. This pipeline operates by: 1) accurately tracking individual identities to extract their mask information and images, and 2) aligning audio segments to each identity through lip synchronization. We curated a dataset of over two million video-entity pairs, capturing both human-human and human-object interactions across a wide range of object categories. In summary, our contributions are as follows:

**(1)** We propose a novel human animation framework capable of synthesizing multi-person and human-object interactions, conditioned on multiple reference images, text descriptions, and audio inputs. The framework also supports long video generation as well as single full-image conditioning. **(2)** We highlight the importance of local condition injection for multi-concept, multi-modal video generation and introduce a simple yet effective design that enables the model to handle both global and local conditions by automatically localizing the conditioned layout. Experimental results demonstrate that our proposed design significantly outperforms existing baselines.

## 2 Related Works

**Video Diffusion Models** have enabled unprecedented quality text-to-video (T2V) or image-to-video (I2V) generation in recent years, thanks to diffusion-based generative models (Ho et al., 2020; Song et al., 2021; Karras et al., 2022; Song et al., 2020; Liu et al., 2022). Early T2V approaches either adapt pretrained text-to-image networks in a training-free manner (Singer et al., 2022; Wu et al., 2023b; Qi et al., 2023) or fine-tune UNet-based latent diffusion architectures (Guo et al., 2023; Zhou et al., 2022; Blattmann et al., 2023a; Wang et al., 2023). To push the frontier, recent works compress spatiotemporal features with 3D causal VAEs (Yu et al., 2023), and migrate to Diffusion Transformer (DiT) backbones (Vaswani et al., 2017; Peebles & Xie, 2023; Brooks et al., 2024; Hong et al., 2022). Through progressive low-to-high resolution pretraining and fine-tuning (Polyak et al., 2024; Kong et al., 2024; Wang et al., 2025a), these models yield longer, more coherent, and high-quality videos. Our method is also built upon the pretrained DiT video generation models.

**Human Animation Models** synthesize videos of people driven by text, reference images, human body poses or audios. Early GAN-based methods (Siarohin et al., 2019; Zhao & Zhang, 2022; Siarohin et al., 2021; Jiang et al., 2024b; Wang et al., 2021) are trained on small datasets (Nagrani et al., 2017; Siarohin et al., 2019; Xie et al., 2022; Zhu et al., 2022) for self-supervised pose transfer. Diffusion-based approaches (Shao et al., 2024; Zhang et al., 2024; Hu, 2024; Wang et al., 2024c; 2025b) now surpass GAN-based methods by conditioning on 2D skeletons, 3D depth, or mesh sequences. Audio-driven portrait methods (Ye et al., 2022; Zhang et al., 2023; Tian et al., 2025b; Jiang et al., 2024a; Cui et al., 2024) have been extended toward full-body motion via two-stage pipelines for improved hand quality (Corona et al., 2024; Meng et al., 2024; Tian et al., 2025a; Hogue et al., 2024) and one-stage unified framework designs (Lin et al., 2024; 2025a). In summary, none of them have explored multi-concept human animation, and our method is the first to enable multi-concept human animation with local audio conditions.

**Multi-Concept Video Customization Models** have received limited attention. Early single-concept identity-preserving methods include Videobooth (Jiang et al., 2024c), which learns coarse-to-fine embeddings from WebVid (Bain et al., 2021) via Grounded-SAM (Kirillov et al., 2023; Liu et al., 2023; Ren et al., 2024); ID-Animator (He et al., 2024), which integrates IP-Adapter (Ye et al., 2023) into AnimateDiff (Guo et al., 2023); and ConsisID (Yuan et al., 2024), which decouples frequency signals to preserve facial identity. More recently, Video-Alchemist (Chen et al., 2025a), ConceptMaster (Huang et al., 2025), and Phantom (Liu et al., 2025) support multiple reference images and text descriptions via cross- or self-attention injection for general-purpose multi-concept customization. BlobGen-Vid (Feng et al., 2025) further enables local layout control of text-specified concepts using user- or LLM-provided spatiotemporal masks. Ingredients (Fei et al., 2025b) proposes to predict the layout of identities in multi-image customization, yet it does not consider the audio condition and does not support multi-person talking generation. All existing methods rely solely on image and text conditions and lack support for multimodal inputs such as audio, which we argue are essential for truly versatile, human-centric video generation.

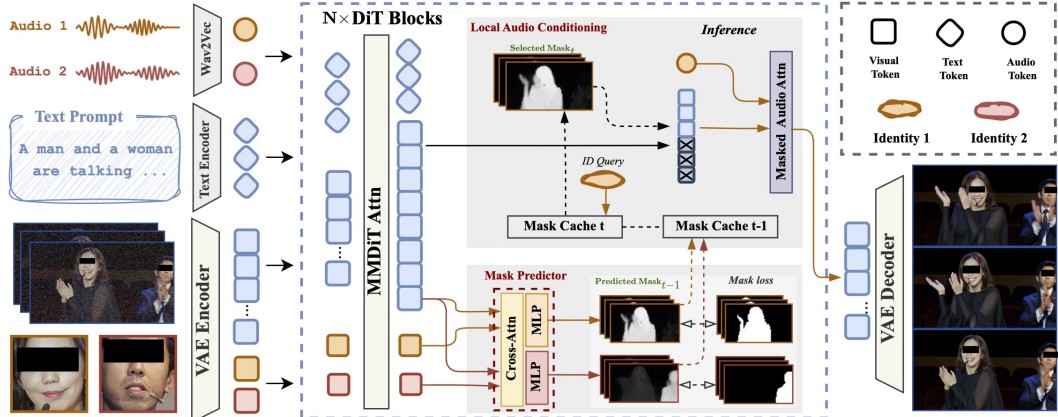

Figure 2: Illustration of our framework, which adaptively predicts masks as the spatial guidance of audio condition injection. In training, we train the mask predictor (cross-attn w/ MLP) with mask loss; in inference, we collect mask predictions to cache and leverage masks predicted from the last denoising step $(t-1)$ to guide the audio cross-attn in the current denoising step $(t)$.

# 3 METHOD

In this section, we present InterActHuman, our multi-concept video generation framework designed to address the challenges of local condition matching for identities in multi-modal conditions. As shown in Fig. 2, the framework begins by customizing reference images of multiple concepts into a video and predicting the mask regions corresponding to each reference appearance in the output video as layout cues with a cross-attention as mask predictor. It operates on the features of the noisy video latent and the reference image latent and supervised by ground-truth masks. Finally, local audio conditions are injected into the designated regions guided by an iterative mask prediction procedure.

## 3.1 PRELIMINARIES

**Problem Setting.** Given a caption $T$ that describes a target video, alongside a set of concept reference images $\{X_i \mid i = 1, \ldots, N\}$ and identity-level audio $\{Y_i \mid i = 1, \ldots, N\}$ where $N$ denotes the number of distinct concepts, the objective of Multi-Concept Human Animation is to generate high-quality videos that faithfully integrate all image-specified visual concepts with correct lip movements synchronized with each audio signal in accordance with the descriptive caption $T$. Each concept should consistently preserve its visual identity as depicted in the provided images, while accurately expressing the semantic roles and behaviors described in the caption.

**Preliminaries.** Transformer-based text-to-video latent diffusion models have recently demonstrated remarkable capabilities in generating high-quality video content. Our proposed InterActHuman framework is built upon the MMDiT-based video generation model (Peebles & Xie, 2023; Esser et al., 2024; Seaweed et al., 2025) and utilizes a 3D Variational Autoencoder (VAE) (Kingma & Welling, 2013), which compresses input videos into a compact latent space. For training, we adopt the flow matching objective (Lipman et al., 2022), which formulates the generative process as a probability flow ordinary differential equation (ODE). This ODE transports clean latent representations $z_0$ to their noisy counterparts $z_t$ along a linear path, defined as $z_t = (1 - t)z_0 + t\epsilon$ at timestep $t$, where $\epsilon$ is sampled from a standard Gaussian distribution. In our setting, which incorporates multi-modal conditions (image and audio), the output of the diffusion transformer is parameterized as $v_\Theta(z_t, t, c_{img}, c_{audio})$. This output is supervised to predict the velocity $(z_1 - z_0)$, resulting in the following training objective:

$$\mathcal{L} = \mathbb{E}_{t,z_0,\epsilon} \left\| v_\Theta(z_t, t, c_{img}, c_{audio}) - (z_1 - z_0) \right\|_2^2.$$

This formulation ensures the model effectively learns to generate videos conditioned on multi-modal inputs while maintaining high fidelity and temporal consistency.

**Multi-Concept Reference Image Injection.** To handle multiple appearance images, we inject appearance cues via self-attention in the original DiT layers. Each reference image $X_i$ is encoded into latent tensors $\mathbf{x}_i$ using the same VAE as for noisy video frames. The reference latents are

then flattened into token sequences and processed by the DiT along with the noisy latents $\mathbf{v}$. The reference latents will reuse the parameters of DiT to extract features and interact with noisy latents at the self-attention layer in every DiT block, allowing appearance cues to propagate to the denoising pathway. Notably, no extra networks or additional parameters are required, thus preserving the model's efficiency. Please refer to Appendix D for more details.

## 3.2 LAYOUT PREDICTION AND LOCAL CONDITION INJECTION

While reference-image injection is well studied, leveraging spatiotemporal layouts for local conditions, such as precise audio alignment, remains challenging. Mask prediction and audio injection are coupled: if a denoising step is incomplete, the masks are unavailable; if it is complete, it is too late to inject local conditions. We resolve this with the multi-step inference of diffusion models (Song et al., 2021): masks predicted at step $k$ guide multi-modal condition injection at step $k+1$. As the network learns spatiotemporal layouts for each identity, precise masks emerge without user annotations.

**Mask Predictor.** In each DiT block, we predict a spatiotemporal mask quantifying how strongly each reference image should influence each video frame. A lightweight mask-predictor head is attached to each of the $L$ transformer layers, consisting of: (1) a shared linear projection that maps hidden video features $\mathbf{h}^v$ and hidden reference features $\mathbf{h}_i^r$ to query, key, and value tensors; (2) LayerNorm; (3) 3D RoPE positional encoding; (4) a cross-attention module; and (5) a two-layer MLP. After normalization and RoPE, video tokens attend to a single reference via $a_i^{(l)} = \mathrm{softmax}\left(\frac{\mathbf{Q}^v \mathbf{K}_i^{r\top}}{\sqrt{d}}\right) \mathbf{V}_i^r$, where $d$ is the head dimension and $l$ indexes the current layer. The attended feature $\mathbf{a}_i^{(l)}$ is transformed by the MLP and passed through a sigmoid to yield a layer-specific mask $\mathbf{m}_i^{(l)} \in [0,1]^T$ for reference $X_i$. We then average the predictions from the last few DiT blocks to obtain the final mask. Notably, the mask predictor is trained to recover the complete human region, regardless of whether the reference image shows only the upper body, face, or full body. This simplifies mask prediction and stabilizes the conditioning module, yielding robust behavior across diverse scenarios. Because the predictors reuse in-block features and share parameters across references, they add minimal overhead while enabling explicit layout control for each identity.

**Mask Prediction by Caching During Inference.** During inference, aggregating mask signals across layers is difficult in early denoising steps, when reference concept latents are still ambiguous. We adopt an iterative strategy: at each step, the mask predicted at the previous step is cached and used as a layout prior for the current step. This progressively sharpens spatiotemporal localization, allowing masks to adapt to the evolving content and remain consistent per concept over time, thereby improving overall conditioning quality.

**Local Audio Conditioning.** To prepare for multi-concept, human-centric video generation, we first pretrain on single-identity audio-conditioned animation by adding cross-attention–based audio conditioning (without masks) and a mixed-conditions training strategy, following (Lin et al., 2025a). Concretely, a new cross-attention layer injects wav2vec (Baevski et al., 2020) audio features into each DiT block after the MMDiT layer. In the multi-concept setting, this pretrained audio cross-attention already provides strong per-identity audio control. We therefore implement *local* audio conditioning primarily at inference: instead of updating all noisy video tokens (Lin et al., 2025a; 2024), we inject wav2vec features only into tokens assigned to a given identity, using the previous-step mask as guidance. To ensure smooth transitions in latent features and in the final video, we blend meaningful and muted audio features per token using the mask confidence, with soft weighting near mask boundaries. This local conditioning also enables multi-person dialogues. Given per-speaker audio tracks as input, the model generates realistic interactions in which speakers take turns, supporting a wide range of applications.

**Training Loss and Strategies.** The overall objective combines a flow-matching diffusion loss (Sec. 3.1) with a focal loss (Lin et al., 2017) for mask classification, which stabilizes training compared to binary cross-entropy, likely due to foreground–background imbalance and occasional low-quality masks. A frame-alignment flag excludes frames with invalid or low-quality masks from loss computation, strengthening temporal supervision. To mitigate the common "copy–paste" behavior of diffusion models—replicating subjects from references with little variation in pose or viewpoint—we randomly mask reference images to reveal only the head, full body, or clothing,

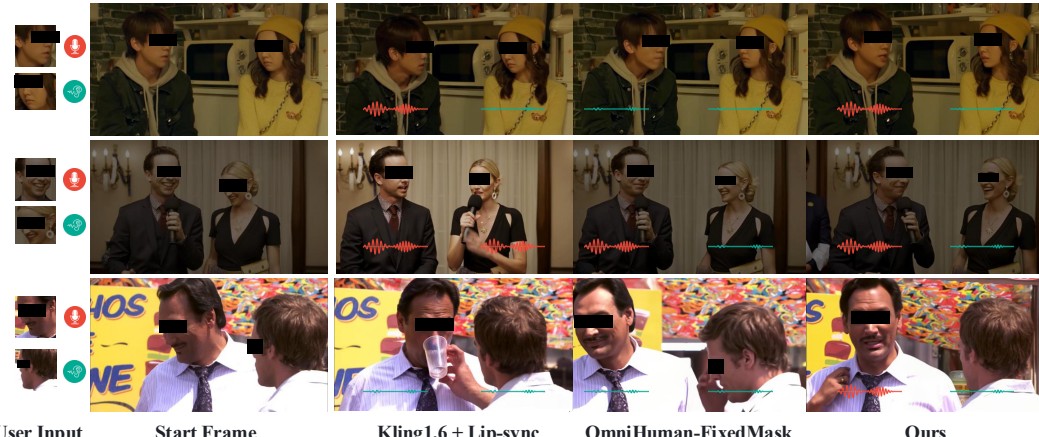

Figure 3: Qualitative comparison with previous methods on multi-concept audio injection.

promoting appearance diversity. We use a face-to–full-body appearance conditioning ratio of $0.7 : 0.3$ and weight the diffusion and focal losses equally $(1 : 1)$.

**Inference Strategies.** Given $N$ reference images and a text prompt, our method uses Qwen2.5-VL (Bai et al., 2025) as a rephraser to extract detailed descriptions from each reference image and integrate them with the original prompt. We apply a shared classifier-free guidance (CFG) across audio and text. For local control, audio conditioning is applied only within the predicted mask regions for each reference. Because early denoising steps yield unreliable masks that may suppress other references, we disable mask-based injection for the first 10 steps, then enable it using the previous-step mask. We inject masked audio only on the conditional (positive) branch during CFG sampling, with a CFG scale of 6.5 and 50 denoising steps.

### 3.3 MULTI-CONCEPT HUMAN-CENTRIC VIDEO DATA CURATION

We curate a large-scale, human-centric video dataset from a large public video dataset (Li et al., 2024) and self-collected videos, filtering out videos that are too short ($<4$ seconds) or contain too many salient humans ($>3$) as detected by a human pose detector (Yang et al., 2023). Unlike previous multi-concept video generation methods (Chen et al., 2025a; Huang et al., 2025) that rely on conventional pipelines like Grounding-DINO (Liu et al., 2023) and SAM (Kirillov et al., 2023) for obtaining a single reference image per identity, our multi-stage pipeline leverages advanced vision-language models. First, each raw video is densely captioned using Qwen2-VL (Wang et al., 2024b) (distilled from Gemini-2.0-Pro), generating fine-grained descriptions of the environment, subjects' appearance, actions, expressions, interacting salient objects, and inter-subject interactions. Next, these descriptions are parsed by a zero-shot Gemini-2.0-Flash API to extract structured appearance phrases. For spatial supervision, we use Grounding-SAM2 (Ren et al., 2024) with the query `person` to produce accurate, temporally consistent masks, which are used both for extracting foreground reference images (with a white background) and as ground-truth for our mask predictor. Our corpus comprises over 2.6M triplets of videos, per-frame masks, and captions, forming the foundation of InterActHuman.

## 4 EXPERIMENTS

**Baselines.** Since ConceptMaster (Huang et al., 2025) and Video-Alchemist (Chen et al., 2025a) are not open-sourced, we do not have access to their models nor test sets. Therefore, we primarily compare our method with recent multi-concept video customization models through their publicly available APIs or public models, evaluating performance from the perspectives of visual appearance and adherence to text prompts, including Vidu2.0 (Bao et al., 2024), Pika2.1 (pik), Kling 1.6 (Kuaishou, 2024) and Phantom (Liu et al., 2025), and audio-driven human video generation methods, including DiffTED (Hogue et al., 2024), DiffGest (Zhu et al., 2023) + Mimiction (Zhang et al., 2024), CyberHost (Lin et al., 2024), OmniHuman (Lin et al., 2025a) and Kling 1.6 (Kuaishou, 2024) with lip synchronization.

Table 1: Quantitative comparisons with audio-conditioned full-body animation baselines.

| Methods | Single-Person Test Set | | | Multi-Person Test Set | | | |
|---|---|---|---|---|---|---|---|
| | Sync-C↑ | HKV↑ | HKC↑ | Sync-D↓ | IQA↑ | AES↑ | FVD↓ |
| DiffTED | 0.926 | - | 0.769 | - | - | - | - |
| DiffGest.+Mimic. | 0.496 | 23.409 | 0.833 | - | - | - | - |
| CyberHost | 6.627 | 24.733 | 0.884 | 8.974 | 4.011 | 2.856 | 54.797 |
| Kling1.6 + Lip-sync. | 4.449 | 46.490 | 0.826 | 8.401 | 4.716 | 3.444 | 33.555 |
| MultiTalk | - | - | - | 7.671 | 4.561 | 3.248 | 35.472 |
| OmniHuman w/o mask | **7.443** | 47.561 | **0.898** | 9.482 | **4.768** | 3.466 | 33.895 |
| OmniHuman w/ fixed mask | - | - | - | 7.068 | 4.690 | 3.369 | 40.239 |
| Ours | 7.272 | **59.635** | 0.885 | **6.670** | 4.757 | **3.467** | **22.881** |

Table 2: User preference evaluation. * means publicly available version with Wan2.1-1.3B.

| Metric | Audio-Driven | | | Multi-Concept Customization | | | | |
|---|---|---|---|---|---|---|---|---|
| | Kling | OmniHuman | Ours | Pika | Phantom* | Kling | Vidu | Ours |
| Avg. Score ↑ | 1.70 | 1.82 | **2.48** | 2.22 | 2.46 | 2.90 | 3.40 | **4.01** |
| Top-1 (%) ↑ | 14.5% | 25.6% | **59.9%** | 4.9% | 9.9% | 13.6% | 22.2% | **49.4%** |

**Evaluation Metrics.** As *audio* and *appearance* are key characteristics of humans, we follow current state-of-the-art methods (Lin et al., 2025a; Huang et al., 2025) to evaluate audio-driven, multi-concept human-centric video generation. To comprehensively evaluate methods for this task, we consider five dimensions: *1) Concept fidelity*: We leverage image feature extractors like CLIP-I (Radford et al., 2021) and DINO-I (Oquab et al., 2023) on subjects in the generated videos to assess whether they are aligned with the provided reference images. To get cropped subject images in output videos, we randomly sample 5 frames in each video and use Florence-2 (Xiao et al., 2024) to detect the subject bounding boxes. We then crop the detected regions and compute the CLIP-I and DINO-I scores. We also employ Face-Arc (Deng et al., 2019), Face-Cur (Huang et al., 2020) and Face-Glink (Deng et al., 2019) to evaluate the fidelity of facial features if the concept is a human. *2) Prompt following*: We utilize video-level CLIP (Wang et al., 2022) to measure the similarity between the input text prompts and the resulting video content. *3) Visual quality*: We utilize q-align (Wu et al., 2023a), a vision-language model, for no-reference image quality assessment (IQA) and aesthetic score estimation (AES). *4) Audio synchronization and human pose diversity*: For lip synchronization, we leverage the widely used Sync-C and Sync-D (Chung & Zisserman, 2017) to compute audio-visual confidence. We also incorporate hand keypoint confidence (HKC) and hand keypoint variance (HKV) (Lin et al., 2024) to quantify hand-pose accuracy and motion richness, respectively. For simplicity, we assume each test video contains exactly two participants: one speaking (with meaningful audio as input) and one listening (with muted audio as input). *5) Distribution distance*: We employ FVD (Unterthiner et al.) to measure the distance between generated and ground-truth videos.

**Test Sets.** We use three test sets in our experiments: 1) single-person audio conditioned human animation test set following OmniHuman (Lin et al., 2025a) (see Tab. 1); 2) our collected two-person audio conditioned human animation test set where only one person is talking (see Tab. 1 and Tab. 4); and 3) multi-concept video customization test set following Phantom (Liu et al., 2025), where we select 100 human-related pairs of reference images and text prompts in our experiments (see Tab. 3).

## 4.1 COMPARISONS WITH STATE-OF-THE-ARTS

**Multi-Concept Audio-Driven Human Video Generation.** In Tab. 1, our approach achieves state-of-the-art or comparable performance in lip synchronization accuracy and motion diversity, particularly excelling in complex multi-person interactions where baseline methods (Lin et al., 2024; Kuaishou, 2024) struggle with accurate audio signal assignments. For single-person scenarios, our method performs on par with specialized models like OmniHuman (Lin et al., 2025a). In multi-person settings, existing methods including OmniHuman and its extensions as well as leading commercial video generation models with post-processed lip-sync, fail to deliver satisfactory results. Poor lip-sync accuracy highlights their inability to generate accurate lip movements for the correct person. Although OmniHuman with oracle masks (manually setting audio-conditioned regions) improves

Table 3: Quantitative comparison of subject consistency, prompt following and visual quality. $^\star$ means publicly available version with Wan2.1-1.3B.

| Methods | Decoupled Concept Fidelity | | | | | Prompt | Video Quality | |
|---|---|---|---|---|---|---|---|---|
| | CLIP-I↑ | DINO-I↑ | Face-Arc↑ | Face-Cur↑ | Face-Glink↑ | ViCLIP-T↑ | AES↑ | IQA↑ |
| Vidu2.0 | 0.696 | 0.458 | 0.568 | 0.562 | 0.597 | 18.61 | 3.350 | 4.689 |
| Pika2.1 | 0.688 | 0.459 | 0.579 | 0.566 | 0.607 | **19.39** | 3.534 | 4.791 |
| Kling1.6 | 0.659 | 0.420 | 0.552 | 0.547 | 0.582 | 18.38 | 3.487 | 4.787 |
| Phantom$^\star$ | 0.703 | 0.476 | 0.589 | 0.573 | 0.615 | 17.73 | 3.404 | 4.812 |
| Ours | **0.744** | **0.533** | **0.598** | **0.600** | **0.644** | 18.87 | **3.565** | **4.903** |

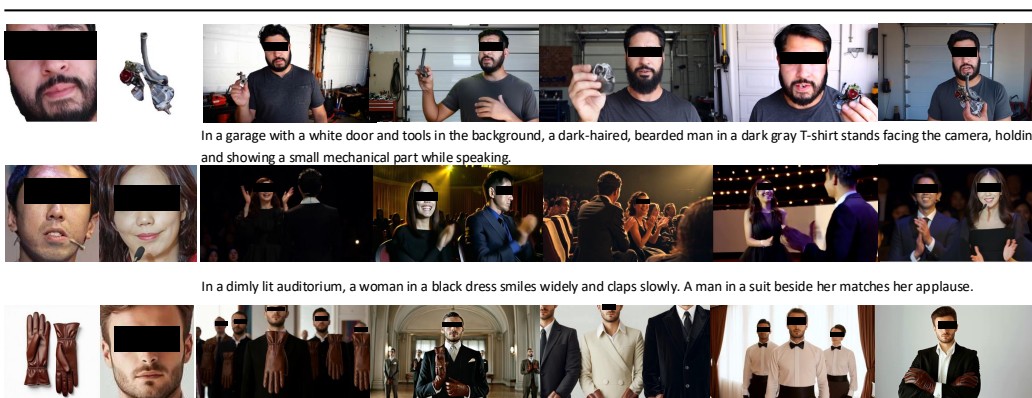

In a garage with a white door and tools in the background, a dark-haired, bearded man in a dark gray T-shirt stands facing the camera, holding and showing a small mechanical part while speaking.

In a dimly lit auditorium, a woman in a black dress smiles widely and claps slowly. A man in a suit beside her matches her applause.

Man, adorned in elegant gloves, stands upright with poised composure, his posture exuding quiet dignity.

**Reference    Vidu2.0    Pika2.1    Kling1.6    Phantom$^\star$    Ours**

Figure 4: Qualitative comparison with previous methods on subject consistency and text following.

lip-sync accuracy, it still significantly degrades overall video quality, as reflected in the FVD metric. This clearly demonstrates the limitations of existing methods that rely on single-identity assumptions, hindering their applicability to broader scenarios. In contrast, our method achieves strong performance in both lip-sync accuracy and overall video synthesis quality, effectively addressing the challenges of audio-conditioned multi-person video generation while remaining compatible with existing settings. Fig. 3 presents qualitative results for multi-person audio-driven video generation. While Kling1.6 w/ lip-sync shows many audio assignment errors and OmniHuman w/ fixed mask shows inflexible audio control with many missing cases, our method consistently assigns audio signals to the correct identity and demonstrates better motion dynamics and more precise audio-driven animation. It is worth noting that all previous methods rely on a reference frame containing complete information to generate talking-person videos, while our method only needs reference appearance of human's head or full-body images and audios.

**User Study.** We conducted a user study to evaluate our method on two tasks: (1) lip synchronization in multi-person talking videos and (2) subject consistency in multi-concept customizations. The lip sync test used 19 videos from three methods, while subject consistency was assessed on 9 videos from five methods. We use the same model (labeled 'ours') for two evaluation tasks. Ten experienced users ranked each method with scores (higher values indicate superior performance). Tab. 2 reports the average scores and top-1 selection percentages. Our method achieved the highest scores and top-1 rates in both tasks, validating the effectiveness of our method.

**Multi-Concept Video Customization.** Although our major contribution is not on multi-concept video customization, we show that our model is also capable of preserving multi-concept visual appearances. In Tab. 3, our method outperforms existing approaches (Bao et al., 2024; pik; Kuaishou, 2024) in preserving identity details and facial features, addressing the common degradation issue in multi-subject generation. Notably, we achieve this without sacrificing audio injection performance, while these methods are incapable of generating videos from audio conditions. It suggests our joint optimization framework on video generation and mask prediction successfully balances video generation. Our method ranks second in prompt following, likely due to our audio-driven human-centric training data focusing on talking and singing, which limits prompt diversity compared to text-to-video tasks. Despite this, qualitative results (Fig. 4) show that our approach maintains natural

Table 4: Ablation study on audio-driven multi-person animation methods.

| Variants | Sync-D↓ | IQA↑ | AES↑ | FVD↓ |
|---|---|---|---|---|
| Global audio condition | 9.482 | **4.768** | 3.466 | 33.895 |
| ID Embedding | 8.627 | 4.658 | 3.338 | 35.665 |
| Fixed Mask | 7.068 | 4.690 | 3.369 | 40.239 |
| Predicted Mask (Ours) | **6.670** | 4.757 | **3.467** | **22.881** |

Table 5: Runtime and parameters versus number of reference images.

| Component | 1 ref | 2 refs | 3 refs | #Params |
|---|---|---|---|---|
| DiT Model | 6.5s | 7.0s | 7.7s | 7B |
| Mask Predictor | 0.4s | 0.8s | 1.2s | 56M |
| Full Model | 6.9s | 7.8s | 8.9s | 7B |

subject consistency and visual quality in both real-world and anime domains, outperforming previous methods that suffer from unnatural compositions and degraded visuals.

## 4.2 ABLATION STUDY

We validate our local audio injection designs via ablation on three variants: *1) Global audio* (Lin et al., 2025a) applies audio across the entire feature map; *2) ID Embedding* injects audio features with a learnable ID embedding without mask prediction; *3) Fixed Mask* uses predefined static spatial masks. As shown in Tab. 4, our framework with predicted dynamic masks achieves the best Sync-D and FVD scores. The fixed mask yields decent Sync-D but suffers motion artifacts (worst FVD), and global audio, while scoring best in IQA, offers poor audio-visual alignment. Qualitative results in Fig. 6 further reveal that global audio drives all identities uniformly, ID embedding often mismatches audio with identities, and fixed masks lose alignment when characters move. These findings underscore that current methods overlook the need for precise local conditions, highlighting the strength of our dynamic, adaptive mask prediction strategy for multi-person talking video generation.

**Computational Cost and Runtime** We measure inference time with the setting of 720p, 109 frames (28 VAE latents) on single A100, and record single forward pass time. The mask predictor adds $\sim$56M parameters versus a 7B DiT and incurs a small overhead per reference image ($\approx$0.013s per DiT block). As more references increase VAE latents from 28 to $28+n$, DiT self-attention scales as $(1+n/28)^2$.

**Failure case analysis of mask prediction** In Fig. 5 we show the qualitative results on the failure case of mask predictions, where the yellow cat is the current identity image (i.e., ID query of mask predictor). We can see that although the mask prediction achieves good performance in IoU measurement from Tables 9 and 10, it still makes mistakes when there are highly overlapping regions. For example, two girls are standing in front of the yellow cat, and the mask predictor manages to exclude the upper body of the right girl in yellow because of small area of overlapping regions. Yet, the mask predictor fails for the girl in pink in both frames. In the first frame, the mask predictor manages to exclude the face region of the little girl in pink, but it mistakenly include another white animal. Besides, due to the high VAE down-sampling ratio of the widely used DiT models, the mask is predicted in a very low resolution, leading to inaccurate mask boundaries.

**Additional analyses in Appendix.** We include (i) a capability matrix against concurrent systems (Tables 8), (ii) mask IoU versus denoising steps/layers under low/high motion plus motion statistics (Tables 9, 10, 11), (iii) a mask cache ablation (Table 12), (iv) scaling to >3 people (Table 13).

## 5 CONCLUSION

In this paper, we introduced InterActHuman, a novel end-to-end video diffusion framework that supports spatially aligned, multi-modal conditioning for multi-concept human animation. By integrating an efficient mask-prediction module into a pretrained DiT backbone, our method automatically infers per-identity spatio-temporal layouts from reference images and uses these masks to guide local audio conditioning. This explicit layout binding enables each reference concept to retain its unique

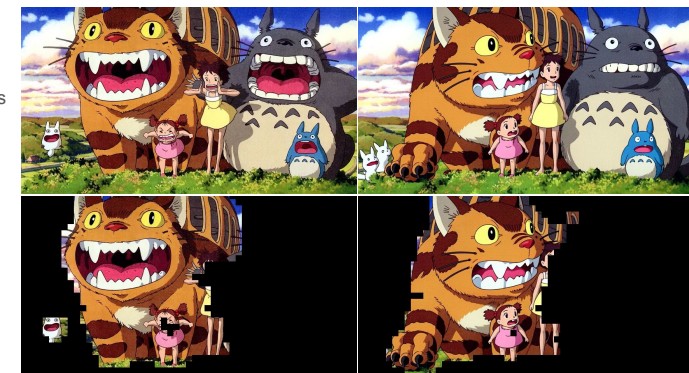

Figure 5: Qualitative results on the failure case of mask predictions.

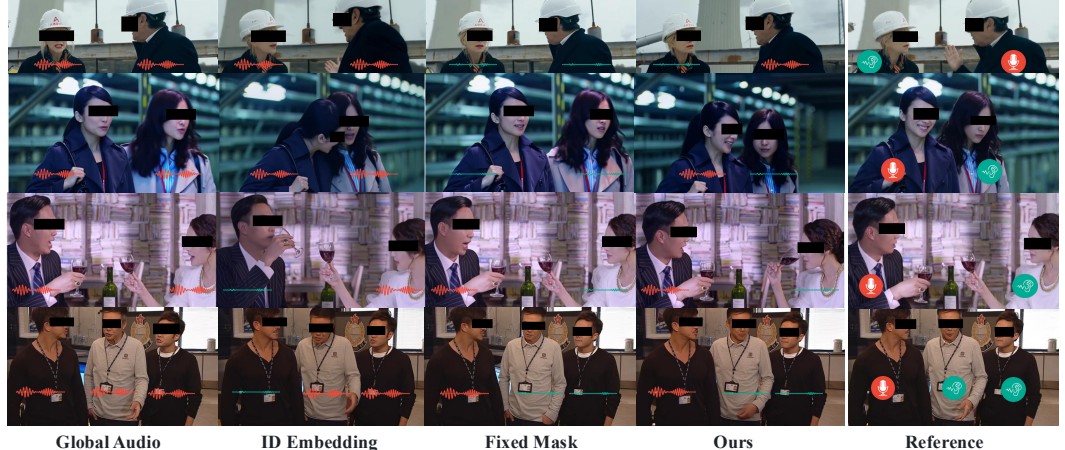

Figure 6: Qualitative ablation on audio injection strategies.

appearance and voice, even in complex scenes with multiple interacting entities. To facilitate training procedure of our model, we presented a high-throughput pipeline for harvesting and annotating over 2.6 million identity-aware video snippets, complete with per-frame masks for humans and objects. Extensive experiments on both single- and multi-person benchmarks demonstrate that InterActHuman achieves state-of-the-art performance in lip synchronization, motion diversity and subject appearance fidelity, while maintaining competitive video quality. We hope it could serve as a solid baseline for the multi-concept human animation and audio-driven multi-person video generation community, where further improvements could be built upon it.

**Limitations.** Since the goal of this work is for human-centric video generation, the available data domain is inherently narrower than that used for general text-to-video pretraining, limiting the ability to follow the diverse text prompts. While our framework is designed to accommodate any number of concept images, the training dataset predominantly consists of videos featuring two to three individuals. This distribution may constrain the generalization of any number of inputs, suggesting further improvements could be achieved by enlarging the diversity and scale of training data.

**Acknowledgment**. We thank the infrastructure team at ByteDance Intelligent Creation for the data curation and annotation support.

**Ethics Statement.** Our model could be used to generate misinformation by using celebrity images and voices. We will strictly restrict access and add watermarks to prevent misuse.

**Reproducibility Statement.** We have provided a code re-implemented on the publicly available video diffusion pretraining model Wan2.1 (Wang et al., 2025a) to show the details of our method. We also provided the dataset processing code, e.g., caption analysis and Grounding-SAM2 (Ren et al., 2024). In the Algorithm 1, we provided detailed pseudo-code to show the inference pipeline of our method. Based on such information, we believe the reproducibility of method is solid enough.

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

# A    ADDITIONAL EXPERIMENTAL RESULTS

## A.1    LLM USAGE STATEMENT.

We use LLMs (e.g., Gemini-2.5 and GPT-5) to polish our paragraphs.

## A.2    COMPARISON WITH SINGLE-PERSON TALKING-HEAD METHODS

Although our method targets multi-person *full-body* talking video generation, we report single-person *talking-head* results for completeness.

Table 6: CelebV-HQ: higher is better for IQA/ASE/Sync-C; lower is better for FID.

| Method | IQA↑ | ASE↑ | Sync-C↑ | FID↓ |
|---|---|---|---|---|
| SadTalker (Zhang et al., 2023) | 2.953 | 1.812 | 3.843 | 36.648 |
| Hallo (Xu et al., 2024a) | 3.505 | 2.262 | 4.130 | 35.961 |
| VExpress (Wang et al., 2024a) | 2.946 | 1.901 | 3.547 | 65.098 |
| EchoMimic (Chen et al., 2024) | 3.307 | 2.128 | 3.136 | 35.373 |
| Hallo-3 (Cui et al., 2024) | 3.451 | 2.257 | 3.933 | 38.481 |
| Loopy (Jiang et al., 2024a) | 3.780 | 2.492 | 4.849 | 33.204 |
| OmniHuman (Lin et al., 2025a) | **3.875** | **2.656** | **5.199** | 31.435 |
| **Ours** | 3.834 | 2.553 | 5.047 | **30.159** |

As shown in Table 6, OmniHuman achieves the best single-person talking-head scores on IQA/ASE/Sync-C, while *our method yields the best FID*. Since our approach targets multi-person, full-body generation rather than single-person heads, these results indicate competitive quality under a setting that is not our primary target. On RAVDESS (Table 7), our method attains the best IQA and ASE, while OmniHuman leads on Sync-C and FID. This mirrors the CelebV-HQ trend and supports our claim that the proposed framework remains competitive on single-person benchmarks while being designed for multi-person, full-body scenarios.

## A.3    COMPARISON WITH CONCURRENT WORKS

We show comparison with recent multi-person talking systems which are conditioned on a *reference frame* only and released concurrently with our method. Our framework supports both concept-only and first-frame modes.

## A.4    MASK REFINEMENT ACROSS DENOISING STEPS AND LAYERS

Across both regimes (Tables 9, 10), IoU steadily improves with denoising steps, validating the *iterative refinement* strategy. Deeper layers (e.g., layer 36) outperform shallow layers early on, while the *combined* mask yields the strongest mid/late-step IoUs. Notably, high-motion sequences retain strong IoUs and achieve competitive lip-sync (Sync-D **6.921**), indicating that masks are *not static* and remain reliable even under larger motion (see the motion statistics in Table 11).

Table 7: RAVDESS: higher is better for IQA/ASE/Sync-C; lower is better for FID.

| Method | IQA↑ | ASE↑ | Sync-C↑ | FID↓ |
|---|---|---|---|---|
| SadTalker (Zhang et al., 2023) | 3.840 | 2.277 | 4.304 | 32.343 |
| Hallo (Xu et al., 2024a) | 4.393 | 2.688 | 4.062 | 19.826 |
| VExpress (Wang et al., 2024a) | 3.690 | 2.331 | 5.001 | 26.736 |
| EchoMimic (Chen et al., 2024) | 4.504 | 2.742 | 3.292 | 21.058 |
| Hallo-3 (Cui et al., 2024) | 4.006 | 2.462 | 4.448 | 28.840 |
| Loopy (Jiang et al., 2024a) | 4.506 | 2.658 | 4.814 | 17.017 |
| OmniHuman (Lin et al., 2025a) | 4.564 | 2.815 | **5.255** | **16.970** |
| **Ours** | **4.602** | **2.915** | 5.132 | 17.187 |

Table 8: Qualitative capability comparison. ✓: supported; **x**: not supported.

| Methods | Audio-driven full-body | First-frame I2V | Multi-ref images | Multi-person talking |
|---|---|---|---|---|
| Video-Alchemist (Chen et al., 2025a) | x | x | ✓ | x |
| ConceptMaster (Huang et al., 2025) | x | x | ✓ | x |
| Phantom (Liu et al., 2025) | x | x | ✓ | x |
| OmniHuman (Lin et al., 2025a) | ✓ | ✓ | x | x |
| HunyuanVideo-Avatar (Chen et al., 2025b) | ✓ | ✓ | x | ✓ |
| MultiTalk (Kong et al., 2025) | ✓ | ✓ | x | ✓ |
| **Ours** | ✓ | ✓ | ✓ | ✓ |

## A.5  EFFECT OF MASK CACHE

Using the cached mask from step $t-1$ to gate audio injection at step $t$ markedly improves lip sync (Table 12; **6.921** vs 11.046 Sync-D). Without caching, identity assignment degrades toward a single-person behavior, confirming the necessity of cross-timestep spatial guidance.

Table 12: Mask cache significantly improves multi-person lip sync.

| Condition | Sync-D↓ |
|---|---|
| With mask cache | **6.921** |
| Without mask cache | 11.046 |

## A.6  SCALING BEYOND THREE PEOPLE

Table 13 shows stable scaling to 4–5 speakers: Sync-D improves slightly (**6.608**) and AES increases (**3.992**) with a marginal IQA change. This aligns with our claim that per-identity mask prediction is independent, enabling more entities without collapsing lip-sync quality.

Table 13: Stable performance when scaling to 4–5 subjects.

| #People | Sync-D↓ | AES↑ | IQA↑ |
|---|---|---|---|
| ≤3 | 6.670 | 3.467 | **4.757** |
| 4–5 | **6.608** | **3.992** | 4.738 |

## B  ALGORITHM OF OUR MODEL IMPLEMENTATION

Please refer to Algorithm 1 for our model implementation.

Table 9: Low motion strength: mask IoU across steps/layers; Sync-D for the setting.

|  | step 1 | step 10 | step 20 | step 30 | step 50 | Sync-D↓ |
|---|---|---|---|---|---|---|
| Mask layer 4 | 0.306 | 0.436 | 0.509 | 0.640 | 0.858 | – |
| Mask layer 20 | 0.386 | 0.629 | 0.733 | 0.890 | **0.957** | – |
| Mask layer 36 | **0.538** | **0.742** | 0.850 | 0.916 | 0.923 | – |
| Combine mask | 0.376 | 0.738 | **0.881** | **0.949** | 0.956 | **7.292** |

Table 10: High motion strength: mask IoU across steps/layers; Sync-D for the setting.

|  | step 1 | step 10 | step 20 | step 30 | step 50 | Sync-D↓ |
|---|---|---|---|---|---|---|
| Mask layer 4 | 0.113 | 0.426 | 0.529 | 0.730 | 0.779 | – |
| Mask layer 20 | 0.527 | 0.816 | 0.911 | 0.934 | **0.945** | – |
| Mask layer 36 | 0.694 | 0.902 | 0.923 | 0.931 | 0.915 | – |
| Combine mask | **0.741** | **0.916** | **0.932** | **0.936** | 0.937 | **6.921** |

## C  MORE DETAILS OF OUR MODEL

### C.1  LONG VIDEO GENERATION AND FRAME-ALIGNMENT DETAILS

We follow sliding-window strategies (Jiang et al., 2024a; Tian et al., 2025b) by reusing several motion frames from the tail of window $w$ as the head of window $w+1$. Reference injection is unchanged; starting frames are optional for window 1.

**Frame alignment flag.**  Per-frame masks have confidence scores from SAM2; frames with confidence $< 0.5$ are excluded from the mask loss (flow-matching still applies). Videos are retained if other frames have reliable masks, so supervision remains temporally dense.

### C.2  LOSS DESIGN AND AUDIO STACK CLARIFICATIONS

**Focal loss.**  We adopt focal loss with $\alpha = 0.25$, $\gamma = 2$ to mitigate class imbalance (background vs person) and stabilize early training; BCE converges more slowly and is less stable initially.

**Audio features.**  Audio tokens come from wav2vec 2.0; we do not add extra transformer blocks before injection. Spatial attention is unrestricted; temporally each latent attends to a local $\pm 5$ token window.

### C.3  IMPLEMENTATION DETAILS

Our model was trained for 10,000 steps on 32 A800 GPUs with a learning rate of 3e-5. We adopted the PyTorch framework combined with Fully Sharded Data Parallel (FSDP) to finetune the DiT model across multiple GPUs. Specifically, the model was partitioned such that different GPUs handled distinct portions of the model's parameters. We configured the effective batch size so that every node—comprising 8 GPUs—processed 2 videos simultaneously. With 32 GPUs in total (i.e., 4 nodes), this resulted in an overall effective batch size of 8 videos.

### C.4  ABLATION DETAILS

In Tab. 4 and Fig. 5 of the paper, we validate our local audio injection designs via ablation on three variants: *1) Global audio* (Lin et al., 2025a) applies audio across the entire feature map; *2) ID Embedding* injects audio features with a learnable ID embedding without mask prediction; *3) Fixed Mask* uses predefined static spatial masks.

For global audio, it is identical to the pretrained audio-driven model (Lin et al., 2025a). For fixed mask, it is a manually input rectangle mask upon the audio cross-attention of pretrained audio-driven

Table 11: Statistics used to define motion regimes.

| Statistic | Low | High |
|---|---|---|
| Body keypoints confidence | 0.737 | 0.663 |
| Max body kp moving dist (px) | 21.268 | 216.269 |
| Var body kp moving dist (px) | 6.098 | 34.022 |
| Head keypoints confidence | 0.9856 | 0.9840 |
| Max head kp moving dist (px) | 2.258 | 34.201 |
| Var head kp moving dist (px) | 2.961 | 216.383 |

model (Lin et al., 2025a). It requires the generated human being static and it needs the user to ensure there is only one person inside a mask.

For ID embedding, here we provide some implementation details. Inspired by detection transformers (Carion et al., 2020), we introduce a set of $N$ learnable ID embeddings $\mathbf{E}_{query} \in \mathbb{R}^{N \times C}$. These embeddings serve as identity tokens for the $N$ individuals. Specifically, we add the same learnable ID embedding $\mathbf{e}_{query} \in \mathbb{R}^{C}$ to a paired reference image and audio segment. In this way, we expect the model to implicitly match an individual (defined by a reference image) and audio segments in the output video to get synchronized lip movements. For the ID embedding dataset preparation, we incorporate multi-person videos into training. We use Active Speaker Detection (ASD) to extract talking persons' identities as a bounding box with identity-number. Then we use the bounding box and SAM2 masks to match the identity-number of each timestep of audio based on their IoU. Finally we use paired audio with timestep and reference image to generate video, and use flow-matching loss only to supervised it. No mask information is provided to the model and no mask supervision is adopted.

In the main paper, our experimental results in ablation study indicate that implicit matching of individuals and audio segments is worse than explicit matching with layout-aligned mask prediction and audio injection. It is worth noting that although our implementation shows this evidence, implicit matching is not necessarily worse than explicit matching. We show that a straight-forward implementation of implicit matching cannot generate satisfactory results, yet there could be better implementations to improve this result in the future.

## D DETAILS OF AUDIO-DRIVEN BASE MODEL'S ARCHITECTURE AND TRAINING

This appendix provides a detailed description of the network structure and training specifics for the pretrained audio-driven single-person video generation model, OmniHuman (Lin et al., 2025a). As noted in our main paper, the text-to-video base model undergoes post-pretraining for audio conditioning, following the OmniHuman (Lin et al., 2025a) methodology. Subsequently, our multi-concept framework is built and trained on this foundation.

### D.1 AUDIO-DRIVEN BASE MODEL

Our foundational model is composed of a Variational Autoencoder (VAE) and a Latent Diffusion Transformer (DiT). The VAE includes an encoder, which compresses raw pixel data into a compact latent representation, and a decoder, which reconstructs the original pixel inputs from these latent features. The VAE achieves compression ratios of (4, 8, 8) for the (t, h, w) dimensions, respectively. Both the encoder and decoder utilize a temporally causal convolutional architecture, facilitating image and video compression across both spatial and temporal domains within the joint latent space. The denoising latents possess 16 channels.

The DiT Blocks are based on the dual-stream Diffusion Transformer (DiT) (Esser et al., 2024). This transformer processes video and text tokens through multiple self-attention layers and feedforward networks (FFNs) to learn representations that are both shared and modality-specific. SwiGLU is employed as the activation function to enhance nonlinear modeling capabilities. Additionally, AdaSingle (Chen et al., 2023) is used for efficient timestep modulation. Moreover, two-thirds of the

FFN weights in the deeper layers share parameters, creating a hybrid-stream design that preserves model capacity while substantially improving parameter efficiency.

## D.2 MIXED TRAINING STRATEGY

We utilize a multi-condition training strategy based on the framework established in OmniHuman (Lin et al., 2025a). Our method employs a two-stage progressive training scheme to develop the base model's capabilities. In the initial stage, the model is trained solely on text-to-video (T2V) data to build fundamental video generation abilities. The second stage introduces audio-synchronized datasets to expand the model's functionality to include audio-driven generation and reference image injection. For this second stage, we follow two core principles from OmniHuman: 1) Tasks with stronger conditioning can leverage tasks with weaker conditioning and their associated data to broaden the effective training dataset; 2) Tasks with stronger conditioning should be assigned proportionally lower training ratios. Guided by these principles, we first train the reference image injection capability before introducing the audio-driven generation objective. This staged methodology facilitates efficient knowledge transfer while ensuring stable training dynamics throughout the multi-condition learning process.

## E ADDITIONAL DETAILS FOR AUDIO-DRIVEN BASE MODEL DATASET

### E.1 DATASET CURATION

We initially filter the T2V data using rules 1 through 6. Following this, we apply rule 7 to further refine the audio-driven data.

**1. Video Clip**. We begin by using PySceneDetect ( https://github.com/Breakthrough/PySceneDetect) to identify and trim shot transitions and fades within video clips. After this process, all clips are standardized to a duration of 5 to 30 seconds.

**2. Human**. Utilizing the annotated video captions, we implement a rule-based system to detect keywords such as "people", "human", "men", "women", "girl", and "boy". If any of these keywords are found, the video is categorized as human-related. This technique ensures the dataset is effectively filtered to comprise only videos pertinent to human activities and interactions.

**3. Subtitles**. We use PaddleOCR ( https://github.com/PaddlePaddle/PaddleOCR) to detect subtitles in the videos and remove clips where subtitles change. This step guarantees that the dataset emphasizes continuous and consistent visual content, reducing distractions from textual variations and improving the data quality for subsequent tasks.

**4. Visual Quality**. We utilize Q-align (Wu et al., 2023a) to evaluate the visual quality of the videos, filtering out clips that do not meet a predefined threshold. This procedure ensures the dataset maintains a high standard of visual clarity, which is essential for accurate analysis and robust model performance. By eliminating low-quality segments, we enhance the overall dependability and utility of the dataset.

**5. Aesthetics**. We employ Q-align (Wu et al., 2023a) to assess the aesthetic appeal of the videos and discard clips falling below a set threshold. Through aesthetic quality assessment, we can filter out videos containing post-production elements, thereby improving the quality of the training dataset. This measure ensures the dataset is composed of natural and unaltered video content, which better represents real-world scenarios and boosts the robustness and generalization of the trained models.

**6. Motion**. We use Raft (Teed & Deng, 2020) to calculate the optical flow of the videos and filter out clips exhibiting overly intense motion. This step ensures that the dataset includes only video clips with moderate and significant motion, which are more appropriate for analysis and model training. By removing clips with extreme motion, we enhance the stability and quality of the dataset, leading to more dependable results.

**7. Syncnet**. For audio-driven content, we employ SyncNet (Chung & Zisserman, 2017) to determine if the lip movements are synchronized with the audio. Videos displaying considerable asynchronization are excluded. This step ensures that the dataset consists only of high-quality, synchronized audio-

visual data. By removing out-of-sync segments, we improve the overall quality and reliability of the dataset. Ultimately, we gather 2,000 hours of audio-driven data.

## E.2 DATASET ANALYSIS

To gain a better understanding of the dataset's distribution, we perform an analysis across three dimensions. Detailed definitions for each dimension are provided below.

- **Human Size**. Human size indicates the portion of the human body visible within the video frame. It is categorized into the following levels: *Portrait* (head and shoulders), *Chest* (from head to chest), *Waist* (from head to waist), *Knees* (from head to knees), and *Full Body* (the entire body is visible). To determine this, we first detect body keypoints using RTMpose (Jiang et al., 2023), and then classify the human size based on the confidence scores of these keypoints. This method ensures a robust and accurate categorization of the visible human body extent in each video.

- **Motion Amplitude**. After extracting body keypoints from the video, the amplitude of human motion is computed by measuring the displacement of the chest keypoint over time, relative to the width of the shoulders. Based on these calculations, we classify motion amplitude into four categories: *Slight*($< 0.1$), *Moderate*($0.1 - 0.2$), *Significant*($0.2 - 0.3$), and *Extreme*($> 0.3$).

- **Scene**. Using video captioning, we employ Doubao to categorize videos into the following scenarios: *Household*, *Work*, *Show*, *Outdoor Adventure*, *transportation*, *Arts and Crafts*, *Sports*, and *Others*.

- **Language**. For data containing audio, we use a language detection tool ( https://github.com/Mimino666/langdetect) to identify and count the types of languages present. These are categorized as *English*, *Chinese*, *Spanish*, *French*, *German*, *Urdu*, *Hindi*, and *Others*.

## F USER STUDY DETAILS

Here we show our questionnaires used in the user study.

### F.1 MULTI-PERSON AUDIO-DRIVEN VIDEO GENERATION QUESTIONNAIRE

Please take 10 minutes to complete the following ranking questions. For each question, consider the priority of the three videos based on the following dimensions and rank them accordingly:

(1) Lip-sync Accuracy: Based on the audio you hear, do you think the lip movements are accurate? Ideally, only one person's lip movements should correspond to the audio, but it doesn't matter specifically who is speaking. It is considered poor if multiple people are speaking simultaneously or if no one is speaking when there is audio. When multiple methods all correctly show only one person speaking, rank them based on the quality of lip-sync with the audio. Perfect synchronization is good; missing syllables or incorrect lip shapes for certain syllables is poor.

(2) Sense of Dialogue Among Multiple People: Is there a feeling of conversation between the individuals? It is considered good if it portrays a natural scenario where one person speaks and another listens. The judgment here is based on whether their expressions during the interaction appear natural and whether the overall feeling of the conversation is natural.

(3) Video Quality: If all the above criteria are tied, then prioritize the video with higher quality.

The priority of these three criteria decreases in the order listed. Aspect ratio and resolution should not be taken into account; you should not favor a particular aspect ratio or higher resolution, but rather focus on the inherent video quality itself.

### F.2 MULTI-CONCEPT VIDEO GENERATION QUESTIONNAIRE

Please take 8 minutes to complete the following ranking questions. For each question, consider the priority of the five videos from the perspective of consistency between the reference images and the appearance in the video, and rank them accordingly:

(1) Consistency with Reference Images: Based on the reference images, do you think all the listed reference images appear in the video? And does the appearance of those in the video match the images? Consider the priority in the following order from high to low:

Ranked highest: Appearance is consistent, and all reference images appear.

Next: All reference images appear, but the appearance of some reference images is inconsistent.

Ranked lowest: Some reference images do not appear.

(2) Video Quality: If the above criteria are tied, then prioritize the video with higher quality.

(3) Motion Strength: If all the above criteria are tied, then prioritize the video with a larger motion strength.

The priority of these criteria decreases in the order listed. Aspect ratio and resolution should not be taken into account; you should not favor a particular aspect ratio or higher resolution, but rather focus on the inherent video quality itself.

---

**Algorithm 1** InterActHuman: Audio-Driven Multi-Concept Video Generation Inference

---

**Require:**
    Text prompt $T$, reference images $\{X_i\}_{i=1}^N$, identity-level audio $\{Y_i\}_{i=1}^N$
    Total diffusion steps $S$, mask injection threshold step $S_{\text{mask}}$
    VAE Encoder/Decoder, diffusion transformer

**Ensure:** Generated video $V$

 1: **Preprocessing:**
 2: **for** $i \leftarrow 1$ **to** $N$ **do**
 3:     $T_i \leftarrow \text{Rephrase}(X_i, T)$                        ▷ Generate detailed prompt via Qwen2.5-VL.
 4:     $\mathbf{x}_i \leftarrow \text{VAE\_Encoder}(X_i)$                       ▷ Encode reference image.
 5:     $\mathbf{a}_i \leftarrow \text{wav2vec}(Y_i)$                             ▷ Extract audio features.
 6: **end for**
 7: $c_{\text{text}} \leftarrow \{T, T_1, \dots, T_N\}$                           ▷ Aggregate text conditions.
 8: $z_S \sim \mathcal{N}(0, I)$                                 ▷ Initialize the noisy video latent.
 9: Initialize mask cache: $\{m_i^{\text{prev}}\} \leftarrow 0$
10: **for** $k \leftarrow S$ **downto** 1 **do**
11:     **for** each DiT block layer $l$ that we inject conditions **do**
12:         **for** $i \leftarrow 1$ **to** $N$ **do**
13:             **Reference Injection:** Inject $\{\mathbf{x}_i\}$ via concatenation and self-attention.
14:             Compute hidden video feature $\mathbf{h}^v$ and reference feature $\mathbf{h}_i^r$.
15:             $\mathbf{Q}^v \leftarrow \text{Proj}_q(\mathbf{h}^v)$.     $[\mathbf{K}_i^r, \mathbf{V}_i^r] \leftarrow \text{Proj}_{k,v}(\mathbf{h}_i^r)$.
16:             Apply LayerNorm and 3D RoPE to the features.
17:             Compute cross-attention:

$$\mathbf{p}_i^{(l)} \leftarrow \text{softmax}\left(\frac{\mathbf{Q}^v(\mathbf{K}_i^r)^\top}{\sqrt{d}}\right)\mathbf{V}_i^r.$$

18:             Predict layer mask:

$$m_i^{(l)} \leftarrow \text{sigmoid}\left(\text{MLP}(\mathbf{p}_i^{(l)})\right).$$

19:         **end for**
20:     **end for**
21:     **Aggregate Masks:**

$$m_i \leftarrow \frac{1}{L}\sum_{l=1}^{L} m_i^{(l)} \quad \forall\, i \in \{1, \dots, N\}.$$

22:     Cache current masks: $\{m_i^{\text{prev}}\} \leftarrow \{m_i\}$.
23:     **if** $k < S_{\text{mask}}$ **then**
24:         **for** $i \leftarrow 1$ **to** $N$ **do**
25:             **Audio Injection**: inject local audio by updating the noisy latent $\mathbf{h}^v$.
26:             $\mathbf{Q}^v \leftarrow \text{Proj}_q(\mathbf{h}^v)$.   $[\mathbf{K}_i^{\text{mute}}, \mathbf{V}_i^{\text{mute}}] \leftarrow \text{Proj}_{k,v}(\mathbf{a}_i^{\text{mute}})$.   $[\mathbf{K}_i, \mathbf{V}_i] \leftarrow \text{Proj}_{k,v}(\mathbf{a}_i)$.
27:             Compute cross-attention:

$$\mathbf{p}_i \leftarrow \text{softmax}\left(\frac{\mathbf{Q}^v(\mathbf{K}_i)^\top}{\sqrt{d}}\right)\mathbf{V}_i, \qquad \mathbf{p}_i^{\text{mute}} \leftarrow \text{softmax}\left(\frac{\mathbf{Q}^v(\mathbf{K}_i^{\text{mute}})^\top}{\sqrt{d}}\right)\mathbf{V}_i^{\text{mute}},$$

28:             Bind audio conditions:

$$\mathbf{h}^v \leftarrow \mathbf{h}^v + m_i \odot \mathbf{p}_i + (1 - m_i) \odot \mathbf{p}_i^{\text{mute}},$$

    where $\odot$ denotes element-wise multiplication.
29:         **end for**
30:     **end if**
31:     Update latent via diffusion step via flow-matching formulas or custom samplers.
32: **end for**
33: **Decoding:** Decode the final latent via VAE:

$$V \leftarrow \text{VAE\_Decoder}(z_0).$$

    **return** $V$

---

