# OpenReview forum: "InterActHuman: Multi-Concept Human Animation with Layout-Aligned Audio Conditions"
_ICLR.cc/2026/Conference — ICLR 2026 Poster_

### Official Review · Reviewer_UaXd · 2025-10-29

**Soundness:** 4
**Presentation:** 4
**Contribution:** 3
**Rating:** 6
**Confidence:** 2

**Summary:**

This paper proposes a human animation framework that synthesizes human-centric videos or images aligned with both audio tracks and text prompts, based on reference images of individuals and multimodal inputs such as audio. Specifically addressing the issue of temporal and identity misalignment between audio tracks and speaking movements of people in the video, the paper introduces a simple yet effective mask prediction mechanism. This mechanism progressively performs local audio condition injection during the denoising process, enabling dynamic audio-spatial alignment without requiring precise masks, thus avoiding incomplete or misaligned generation.

**Strengths:**

1. Per-identity control of multiple concepts is an urgent problem in human animation tasks; this paper offers a simple yet reliable solution that is not limited by the number of people in the video and shows strong scalability.
2. The local-audio-conditioning injection strategy is both novel and straightforward, and its effectiveness is verified by experiments and user studies.
3. The inference results are excellent; the supplementary videos are particularly impressive.
4. The paper explains its core contributions clearly, concisely, and without ambiguity.

**Weaknesses:**

1. Owing to the limited scope of the training data, the method’s ability to follow detailed instructions appears constrained; it is unclear whether this stems from a fundamental limitation in the approach or simply from the narrow range of scenarios covered by the dataset.
2. The generated humans exhibit rather subdued body motion. Yet in real conversations, lively gestures are equally important. This could be a valuable direction for the community to explore in the future.

**Questions:**

The paper focuses on audio and image as the multimodal conditioning cue. Could the same framework, for instance, be adapted to accept additional signals, such as skeletal key-points, so that the generated humans would perform richer, visually matched motions?

---

> ### Author Response · Authors · 2025-11-20
>
> Thank you for the constructive review. We appreciate your recognition that our problem setting is urgent and that our method is both novel and straightforward. Below, we address the main points.
>
> **Q1: Instruction following under limited training scope**
>
> A1: Thank you for highlighting prompt adherence. This concern is common in human image animation and largely reflects a known trade-off when fine-tuning large pretrained models: while fine-tuning adds new capabilities, it can also attenuate some original ones (i.e., partial forgetting). We are actively working to better preserve and improve text-prompt following during task-specific training and will continue to report progress on this front.
>
> **Q2: Importance of lively gestures**
>
> A2: We agree that emotion, expressiveness, and gesture dynamics are crucial in multi-person interaction. Our current work focuses on identity preservation and audio-synchronized talking. We hope it can serve as a foundation for future efforts that place greater emphasis on rich, lifelike gestures and affect.
>
> **Q3: Extending the framework with skeletal keypoints**
>
> A3: In principle, incorporating skeleton keypoints is feasible. A practical challenge arises with pixel-aligned conditions (e.g., per-pixel keypoint maps), which our framework does not directly utilize. However, pixel-agnostic signals, such as expression or full-body latent features embedded in a shared semantic space, are compatible and can be integrated alongside our existing audio features. We view this as a promising direction for future extensions.
>
> We hope these clarifications are helpful, and we appreciate the insightful feedback. Thanks for your time to review our paper!

---

> > ### Comment · Reviewer_UaXd · 2025-11-28
> >
> > Thank you for your response. I am particularly interested in experiments combining audio features with other features. If you could include some relevant experimental demonstrations in the paper, it would greatly enhance the multimodal scalability of the presented method. For now, I will maintain my rating.

---

### Official Review · Reviewer_xdLa · 2025-10-30

**Soundness:** 3
**Presentation:** 2
**Contribution:** 2
**Rating:** 4
**Confidence:** 4

**Summary:**

The paper proposes InterActHuman, a diffusion transformer based human video generation framework that aims to synthesize multi person or human object interaction videos in which each identity keeps its own appearance, motion style, and voice, given multiple reference images, a text description, and per speaker audio tracks. The core idea is to drop the usual single identity assumption and instead bind each conditioning signal to the correct spatial region over time. To do this, the model predicts for each reference concept a spatiotemporal mask via a lightweight cross attention head that matches the noisy latent video tokens to that concept. These masks are refined iteratively across denoising steps and then used to inject audio features only into the region of the speaker at the next step, which addresses the chicken and egg problem of needing masks to apply localized audio while the video is still being generated.

**Strengths:**

1. The paper tackles a practically important and under explored setting: multi-person, audio-conditioned human animation where each identity must keep its own appearance and voice, instead of the common single identity assumption in prior audio-driven portrait or OmniHuman style models.

2. The proposed iterative mask prediction and cached layout guided audio injection mechanism is elegant. By predicting per identity spatiotemporal masks using cross attention between reference appearance tokens and noisy video latents, then using the previous step mask to gate the next step audio cross attention, the model effectively solves the chicken and egg problem of aligning local audio before the final frames exist.

3. Quantitative and qualitative results are compelling. The user study shows a strong preference for the proposed method in both lip sync realism and subject consistency.

**Weaknesses:**

1. Runtime cost and scalability claims are mostly deferred to the appendix. The main text asserts minimal overhead and compatibility with long video generation, but does not quantify inference speed or memory usage when conditioning on multiple identities.

2. Multi speaker audio assignment appears to rely on injecting each audio stream only into the spatial region indicated by that speaker’s cached mask at inference time. It is unclear whether the model is ever explicitly trained on multi speaker scenes with multiple simultaneous audio streams, or whether this is essentially a zero shot composition of single speaker training. This could limit robustness when speakers interrupt each other or overlap.

**Questions:**

1. How are individual audio streams associated with the correct identity during training. The paper explains that during inference, audio cross attention is only applied to tokens whose mask corresponds to that speaker, using cached masks from the previous denoising step.

2. The data pipeline aligns audio segments to identities through lip synchronization and produces per-frame masks using Grounding SAM2 with a person query. Could you elaborate on how you ensure temporal identity consistency across frames in multi-person scenes, especially during occlusion or when two people have similar appearances? Do you rely on optical flow tracking or an ID re-identification module?

---

> ### Author Response · Authors · 2025-11-20
>
> Thank you for the constructive review. We appreciate your recognition that our problem setting is practically important yet under-explored, and that our proposed approach is elegant. Below we address the main points.
>
> **Q1: Quantifying inference speed and memory usage with multiple identities**
>
> A1: Thank you for the suggestion. We now report inference memory usage and parameter changes when conditioning on multiple identities in the main text (moved from Table 13 in the Appendix), alongside clarifying notes on compute. Please see the updated Experiments section for details (Line 456-459).
>
> **Q2: Multi-speaker training and association of audio streams with identities**
>
> A2: We appreciate this question. In our formulation, we assume at most one active speaker at any given instant. When multiple people talk simultaneously, current models including ours can only make all identities speak the same content (i.e., feed the same audio to all identities). To the best of our knowledge, no works could process multiple audio signals at the same time without audio source separation (i.e., take a multi-person discussion audio as input, where many people talking simultaneously). With additional audio source separation models, this problem could be converted to the existing problems our model could handle, i.e., separated audio signals for each identity. It could also work even when the audios have temporal overlaps.
>
> For training on multi-speaker scenes, we use Active Speaker Detection (ASD) to associate extracted human masks with corresponding audio segments. In practice, we found the accuracy of current ASD methods to be limited, which reduces the benefit of explicitly training with overlapping speech. Empirically, we observed similar Sync-D performance for the following two variants:
>
> *Variant 1*: Single-person audio pretraining + multi-person visual training: train audio cross-attention on a single-person talking dataset, and then train multi-image customization and the mask predictor on multi-person data. Don't train audio cross-attention on multi-person videos because of limited ASD accuracy.
>
> *Variant 2*: Joint training of every trainable module on multi-person data: pretrain audio cross-attention on single-person data, then further train audio cross-attention together with multi-image customization and the mask predictor on multi-person data.
>
> Given the comparable results, we adopt Variant (1) in the paper and report its performance. At inference, masks are predicted autoregressively (cached from the previous step) rather than provided as inputs. Thus, the mask predictor and multi-reference image-to-video components are always trained on multi-person data, while further training audio cross-attention on multi-person data yielded only marginal differences in our tests. We acknowledge that robust handling of simultaneous speech remains an open direction and future works could revisit this with stronger ASD or explicit overlap supervision.
>
> **Q3: Temporal identity consistency under occlusion or similar appearances**
>
> A3: Our training clips are intentionally short (about 5–10 s), which limits the need for long-term tracking. We do not rely on optical-flow tracking or a re-identification module. Instead, we emphasize data curation for temporal consistency: we validate SAM2-based tracklets and compare the number of tracklets with person counts estimated by pose detectors. If a clip with 2–3 people (ensured by pose detection) yields fragmented tracklets from SAM2 (e.g., >2–3), suggesting occlusion or difficult cases, we discard it. This filtering reduces severe occlusion scenarios in training while maintaining consistent identity trajectories within clips.
>
> We hope these clarifications are helpful, and we appreciate the insightful feedback.

---

### Official Review · Reviewer_SmPL · 2025-10-31

**Soundness:** 3
**Presentation:** 2
**Contribution:** 2
**Rating:** 4
**Confidence:** 5

**Summary:**

This paper proposes InterActHuman, a video diffusion framework for multi-concept human animation, incorporating layout-aligned multimodal conditions such as local audio and image references for distinct entities. The approach introduces a mask-predictor module, iteratively integrated into the denoising steps of a diffusion transformer, to spatially align conditioning signals (audio/image) with their corresponding regions. Experiments and ablations on curated large-scale datasets reportedly demonstrate state-of-the-art performance in multi-person talking and human-object interaction video generation.

**Strengths:**

1. The paper provides relatively comprehensive quantitative evaluations using mainstream avatar-related metrics.
2. The authors offer detailed descriptions of how they collected and cleaned a large-scale dataset combining reference images, audio, per-frame masks, and captions for diverse multi-human/object interactions (Section 3.3), which could be a valuable resource for empirical studies if released publicly.

**Weaknesses:**

1. While using layout-based guidance for multi-concept conditioning is a reasonable design, the claimed novelty of this framework is questionable. Similar strategies are already standard in both multi-concept image and video generation. Even within avatar-related tasks, several prior works, including MultiTalk[1], have adopted comparable layout-guided conditioning. Moreover, dynamic layout prediction was first introduced in Ingredients[2], where it serves a clear purpose in ipt2v task. However, for avatar generation tasks, the spatial configuration of characters is typically known beforehand. In such cases, slightly loosening pre-defined masks can often achieve sufficient flexibility without introducing additional model complexity. The paper does not convincingly demonstrate that the proposed learnable mask predictor provides meaningful gains over these simpler, training-free alternatives.
2. The experimental evaluation relies entirely on a self-collected, closed-source test set. This choice severely limits reproducibility and undermines the credibility of the reported improvements. Without standardized or publicly available benchmarks, it is difficult to assess whether the observed gains generalize beyond the authors’ specific data.
3. The paper does not specify which base video diffusion model was used for initialization, nor does it provide details about training infrastructure (e.g., number of GPUs, total data scale, or training duration). More importantly, it remains unclear whether the base model natively supports audio conditioning or if such capability was newly introduced by the authors. Without this information, it is difficult to assess how much of the reported performance stems from the proposed mechanism itself versus the underlying backbone or large-scale training setup. Given that the paper reports notably strong results, this omission raises concerns about the fairness and interpretability of the comparisons. A more transparent description of the training setup and backbone configuration is essential to substantiate the claimed contributions.
[1]. Let Them Talk: Audio-Driven Multi-Person Conversational Video Generation.
[2]. Ingredients: Blending Custom Photos with Video Diffusion Transformers.

**Questions:**

In Line 228-229, what is the meaning of '!'; besides, 'the last few DiT blocks' should be specified to be reproduced.

---

> ### Author Response · Authors · 2025-11-20
>
> Thanks for the constructive review. We appreciate the acknowledgment that our model substantially outperforms prior SOTA. Below we address the main points.
>
> **Q1: Novelty, comparison with MultiTalk, Ingredients and training-free alternatives.**
>
> A1: Firstly, we want to express that our work was conducted concurrently with MultiTalk [1]. We have reported quantitative comparisons in Table 1 of our original paper. We also highlight a key functional difference with MultiTalk in Table 8: MultiTalk does not accept multiple reference images as input, which is central to our problem setting. Therefore, our model generalizes better in the image condition part than MultiTalk because we further takes multiple reference images while MultiTalk is the traiditional first-frame to video generation setting (I2V).  Our method was also developed independently of Ingredients [2]. As it does not consider audio conditioning, its contribution is largely different from ours. Even it also predicts layouts, the purpose of its layout prediction is significantly different from ours. Because our primary contribution is multi-person talking generation, the existence of Ingredients does not diminish our novelty. We have added the appropriate citation and discussion in the updated manuscript (line 150).
>
> Regarding the claim that “avatar generation typically has a known spatial configuration,” this does not apply to our setup. When the inputs are multiple reference images (often two headshots/person images) and no complete image conditions like first-frame are provided, there is no pre-specified layout. Our input is significantly different from the traditional I2V setting used by MultiTalk, where multi-person spatial layout is known.
>
> Finally, we evaluated a training-free alternative that loosens predefined masks. As reported in Table 4, this baseline (a single-person talking method with predefined masks) underperforms, whereas our learnable mask predictor yields consistent gains, supporting its value.
>
> **Q2: Public test set**
>
> A2: To our knowledge, there is currently no public benchmark tailored to multi-person talking scenarios, which motivated our self-collected test set. We mitigate this limitation by detailing our data collection and evaluation protocols to facilitate reproducibility, and we welcome pointers to existing public benchmarks to incorporate in evaluations.
>
> **Q3: Details of the pretrained model**
>
> A3: We initialize from an internal 7B MMDiT-based text-to-video backbone, whose details could be found in [a]. The base model does not natively support audio. We introduce an audio cross-attention module and first continue training on a self-collected single-person talking dataset before applying the multi-person training in the paper (see Appendix §D). For fairness and interpretability, we report extensive ablations against a model trained only on the single-person dataset with the same backbone and compute, showing that improvements stem from our multi-person design rather than scale.
>
> [a] Seawead, Team, Ceyuan Yang, Zhijie Lin, Yang Zhao, Shanchuan Lin, Zhibei Ma, Haoyuan Guo et al. "Seaweed-7b: Cost-effective training of video generation foundation model." arXiv preprint arXiv:2504.08685 (2025).
>
> **Q4: Typos and mask injection in the last DiT blocks**
>
> A4: We corrected the formula typo in the updated paper. For mask injection, we average the masks from layers 20, 24, 28, 32, and 36 by default. Please see Appendix §A.4 for additional ablations on deriving masks from different layers.
>
> We hope these clarifications are helpful, and we appreciate the insightful feedback. Should you have another further questions, please be free to add comments. Thanks for your time to review our paper!

---

### Official Review · Reviewer_6miK · 2025-11-03

**Soundness:** 3
**Presentation:** 3
**Contribution:** 3
**Rating:** 6
**Confidence:** 4

**Summary:**

This paper presents InterActHuman, a novel video diffusion framework for multi-concept human animation, addressing the limitations of existing global conditioning methods in multi-person scenarios. The key contribution is a method for enforcing strong, region-specific binding of local conditions, particularly aligning audio to the correct speaker. The model integrates a mask predictor into the diffusion pipeline to explicitly infer the spatiotemporal layout for each reference identity. It solves the problem of mask prediction during inference by adopting an iterative strategy, where the mask predicted at one step guides the local audio injection at the next. The authors also contribute a large-scale dataset of over 2.6 million video-entity pairs to facilitate this task.

**Strengths:**

1: The framework introduces the capability for multi-person, audio-driven animation, correctly assigning distinct audio streams to specific individuals in the generated video.

2: The paper proposes a practical iterative mask-caching strategy to solve the "chicken-and-egg" problem of local conditioning, using the mask predicted at step $k$ to guide the local audio injection at step $k+1$.

3: Experiments show that the method significantly outperforms existing baselines (like Kling 1.6 w/ lip-sync and OmniHuman w/ fixed mask) on multi-person benchmarks, achieving state-of-the-art lip-sync (Sync-D) and video quality (FVD) metrics.

**Weaknesses:**

1: **Over-reliance on a Private, Curated Dataset.** The model is trained and evaluated on a new, large-scale dataset (2.6M pairs) curated by the authors. The multi-person test set is also newly collected by the authors. This makes it difficult to assess robustness and generalizability. Since the baselines were not trained on this specific, mask-annotated dataset, it's unclear if the performance gap is due to the model's architecture or its specialized training data, which may be perfectly tailored to its design.

2: **Missing Ablation on Inference Hyperparameters.** The core inference strategy to solve the "chicken-and-egg" problem involves disabling mask-based injection for the "first 10 steps" because early masks are deemed "unreliable". This 10-step threshold is a critical, unexplained hyperparameter. The paper provides no ablation study to justify this specific number or to analyze the sensitivity of the model's performance (e.g., in Sync-D or FVD) to this "warm-up" period.

3: **Lack of Failure Case Analysis for Mask Prediction.** The entire method for local conditioning hinges on the accuracy of the predicted mask. While the paper shows average IoU improving over denoising steps (Tables 8 & 9), it provides no qualitative analysis of failure cases.

Overall, this paper proposes a new framework to generate multi-person videos with audio and has comprehensive experiments. Thus, I vote for acceptance at this time.

**Questions:**

1: What does "ID query" in Figure 2 mean? Is it the tokens of ID 1?

---

> ### Author Response · Authors · 2025-11-20
>
> Thanks for the thoughtful review. We’re encouraged that you found our model substantially outperforms prior SOTA. Below we address the main concerns point-by-point.
>
> **Q1: Over-reliance on a private, curated dataset**
>
> A1: Because this domain lacks public training data, we have to curate our own dataset for this work. Our 2.6M-clip corpus is derived from the publicly available OpenHumanVid [a] dataset (13M clips) together with self-collected video data, using the procedures in Appendix §E.1 for both single- and multi-person videos. For multi-person cases, we additionally apply the pipeline described in §3.3.
>
> While policy restrictions prevent releasing the curated videos themselves in our project, we will release all data-processing code and scripts needed to reproduce our training set. By following the OpenHumanVid download instructions and running our scripts, everyone with GPU resources can reproduce >1.5M high-quality human-centric videos (with synchronized audio and associated human-centric annotations) from the original 13M clips. We will also release our test set to enable fair comparisons going forward.
>
> [a] Li, Hui, et al. OpenHumanVid: A large-scale high-quality dataset for enhancing human-centric video generation. Proc. CVPR, 2025.
>
> **Q2: Ablation on the step number for injecting masked audio features**
>
> A2: We performed an ablation on the diffusion step after which we inject audio features using the predicted mask. Injecting after 10 steps yields the best score among the tested settings. Generally, our model is robust to this hyper-parameter as long as it is small (e.g., 5 or 10 when the total is 50) to let the mask predictor be effective for the rest of steps.
>
> | Metric \ Steps |     5 |    10 |    15 |    20 |    30 |
> | -------------- | ----: | ----: | ----: | ----: | ----: |
> | **Sync-D**     | 6.787 | 6.670 | 7.197 | 7.051 | 7.050 |
>
>
> **Q3: Failure case analysis for mask prediction**
>
> A3: We added failure cases in Figure 5 of the updated paper (please refer to the updated pdf file) and more analysis in Line 461-469. The primary failure mode is under heavy occlusion/overlap, where the mask predictor occasionally includes the occluder. For example, when a group stands in front of the target identity (a yellow cat), the predicted mask may partially include the people. Since our model’s core objective is generation, such difficult perception errors (e.g., overlapping instance segmentation) can occur. Importantly, when identities are well separated, performance is strong, as reflected by the mask IoU results in Table 8 and 9 (in the updated paper pdf, it is now Table 9 and 10).
>
> **Q4: Clarification of “ID query” in Figure 2**
>
> “ID query” denotes the mask currently being predicted. In Figure 2, the shown mask corresponds to the woman (assigned ID 1 in the audio-image inputs on the left). In practice, masks for all IDs are predicted in a batch, and each predicted mask is then paired with its corresponding ID’s audio features when injected into the DiT.
>
> Should you have another further questions, please be free to add comments. Thanks for your time to review our paper!

---

### Meta-Review · Area_Chair_DoVc · 2025-12-24

**Summary:**

The main concerns are about the proprietary dataset and the training details. As to the proprietary dataset, the authors promised to release all data-processing code for curating the training dataset, and a test dataset for evaluation. Despite the great efforts, the authors are encouraged to release the dataset derived from OpenHumanVid for the community. Meanwhile, the authors  clarified the specifics on failure case, backbone, training steps, occlusion cases and so on, which is clear and sound. Taking all into account, I am inclined to accept the paper.

**Reviewer Concerns:**

I believe the authors have addressed the reviewers' concerns about the proprietary dataset and the training details. Meanwhile, the authors are encouraged to release the dataset derived from OpenHumanVid considering per-id control for multi-person scenario is an emerging topic.

**Reviewer Scores:**

The average score is 5.0, which I think is proper. I believe most reviewers will be satisfied with the rebuttal and probably maintain their scores.

---

### Decision · Program_Chairs · 2026-01-26

Accept (Poster)